# NESTED LEARNING FOR MULTI-GRANULAR TASKS

## ABSTRACT

Standard deep neural networks (DNNs) used for classification are trained in an end-to-end fashion for very specific tasks - object recognition, face identification, character recognition, etc. This specificity often leads to overconfident models that generalize poorly to samples that are not from the original training distribution. Moreover, they do not allow to leverage information from heterogeneously annotated data, where for example, labels may be provided with different levels of granularity. Finally, standard DNNs do not produce results with simultaneous different levels of confidence for different levels of detail, they are most commonly an all or nothing approach. To address these challenges, we introduce the problem of *nested learning*: how to obtain a hierarchical representation of the input such that a coarse label can be extracted first, and sequentially refine this representation to obtain successively refined predictions, all of them with the corresponding confidence. We explicitly enforce this behaviour by creating a sequence of nested information bottlenecks. Looking at the problem of nested learning from an information theory perspective, we design a network topology with two important properties. First, a sequence of low dimensional (nested) feature embeddings are enforced. Then we show how the explicit combination of nested outputs can improve both robustness and finer predictions. Experimental results on CIFAR-10, MNIST, and FASHION-MNIST demonstrate that nested learning outperforms the same network trained in the standard end-to-end fashion. Since the network can be naturally trained with mixed data labeled at different levels of nested details, we also study what is the most efficient way of annotating data, when a fixed training budget is given and the cost of labels increases with the levels in the nested hierarchy.

## 1 INTRODUCTION

Deep learning is providing remarkable computational tools for the automatic analysis and understanding of complex high-dimensional problems [Esteva et al. (2017); Spanhol et al. (2016); Parkhi et al. (2015)]. Despite its tremendous value and versatility, methods based on Deep Neural Networks (DNNs) tend to be overconfident about their predictions and limited to the task and data they have been trained on [Hein et al. (2018); Guo et al. (2017); Nguyen et al. (2014)]. This happens, among other reasons, because the standard approach to train DNN models consists in optimizing its performance over a specific dataset and for a specific task in an end-to-end fashion Szegedy et al. (2014). Standard DNNs are not designed to be trained with data of different quality and to simultaneously provide results at multiple granularities.

Take as an example the case illustrated in Figure 1 (left); for high quality facial images, we may be able to infer the person's age group and identity; whereas for poor resolution or occluded examples, only a sub-set of these nested predictions may be achievable. We expect the network to automatically understand what can and cannot predict, and this is obtained with the framework proposed in this paper (Figure 1, right). Moreover, nested learning allows us to leverage training information from diverse datasets, with varying granularity and quality of labels, and combine this information into a single model.

On the other hand, when heterogeneous data with different quality and granularity of annotations (as in the example illustrated in Figure 1 (left)) is provided for training, low quality samples with coarse labels can help us to understand the structure of the coarser distributions (person, under 50) while simultaneously data with finer labels can contribute to the coarse and fine tasks. This will be formalized later in the paper with tools from information theory.

Figure 1: On the left, an illustration of a set of nested predictions and their associated confidence given an input image of a face. The top block illustrates a desired behavior. Depending on the quality of the input data, one may be able to provide up to a certain level of prediction. This *nested learning* is the problem addressed in this paper. The bottom block illustrates how standard DNN-based models behave when they are trained in a end-to-end fashion to perform specific tasks such as face recognition. Clearly the traditional network is over-confident in its predictions (potentially wrong for the last 3 cases), and provides an all or nothing response instead of responding *only* what it can for the given input quality. On the right, we see a real example with results from the proposed framework; while a sharp image gets all the nested levels with high confidence in our proposed system, a low-quality one is getting the first two levels with confidence, while the finer one is correct but low confidence as expected. (Additional examples are presented in Figure 5 in the supplementary material.)

Recently, Alsallakh et al. (2017) showed that convolutional neural networks (CNNs) naturally tend to learn hierarchical high-level features that discriminate groups of classes in the early layers, while the deeper layers develop more specialized feature detectors. We explicitly enforce this behaviour by creating a sequence of nested information bottlenecks. Looking at the problem of nested learning from an information theory perspective, we design a network topology with two important properties. First, a sequence of low dimensional (nested) feature embeddings are enforced for each level in the taxonomy of the labels. This encourages generalization by forcing information bottlenecks [Tishby et al. (1999), Shwartz-Ziv & Tishby (2017)]. Second, skipped connections allow finer embeddings to access information of the input that may be useful for finer classification but not informative on coarser categories Ronneberger et al. (2015). Additionally, we show how the explicit calibration and combination of nested outputs can improve the finer predictions and improve robustness. Finally, having the flexibility of merging data with different levels of granularity inspired us to study which is the most efficient way of annotating data given a fixed budget that takes into account that more detailed training data is more expensive to annotate. The source code associated to this work is open source.[1]

The main contributions of this paper are: (1) We introduce the concept of explicit nested learning, where a given level in the hierarchy strictly contains the previous one and strictly adds information; (2) We provide a deep learning architecture that can be trained with data from all levels of the nested structure (all levels of labels and data quality), each one affecting the corresponding component of the network; (3) We provide a model with multiple outputs, one per level of the nested hierarchy, each one with its own confidence; the user does not need to know the "quality" of the data beforehand, the output confidences provide that information.

## 2 Related Work

The problem of adapting DNNs models and training protocols to encourage nested learning shares similarities with other popular problems in machine learning such as Multi-Task Learning (MTL). Though our work is related to MTL because of the similar training challenges, most of the MTL methods tackle the task in a parallel way, without imposing a task hierarchy in the architecture [Ranjan et al. (2016), Kokkinos (2017), Bilen & Vedaldi (2016)]. The idea of learning hierarchical representations to improve classification performance has been exploited prior the proliferation of DNNs, e.g., [Zweig & Weinshall (2007), Fergus et al. (2010), Zhao et al. (2011), Liu et al. (2013)]. Some of these ideas have been incorporated into deep learning methods [Alsallakh et al. (2017), Wehrmann et al. (2018), Deng et al. (2014), Srivastava & Salakhutdinov (2013)], and exploited in specific applications [Clark et al. (2017), Xuehong Mao et al. (2016), Seo & shik Shin (2019)].

---

[1]https://github.com/nestedlearning2019

Kim et al. (2018) proposed a nested sparse network architecture with the emphasis on having a resource-aware versatile architecture to meet (simultaneously) diverse resource requirements. Wehrmann et al. (2018) proposed a neural network architecture capable of simultaneously optimizing local and global loss functions to exploit local and global information while penalizing hierarchical violations. Triguero & Vens (2016) investigated different alternatives to label hierarchical multi-label problems by selecting one or multiple thresholds to map output scores to hierarchical predictions, focusing on performance measures such as the H-loss, HMC-loss and the micro-averaged F-measure. Yan et al. (2015) introduced hierarchical deep CNNs (HD-CNNs) which consists of embedding CNNs into a two-level category hierarchy. They propose to distinguish a coarse class using an initial classifier and then refine the classification into a second level for each individual coarse category.

Although the works listed above are important, relevant, and related to the work presented here, there are notable differences between them and what we propose. For example, while Kim et al. (2018) propose a nested architecture providing different (potentially nested) outputs, they do not study how to combine these outputs into a refined single prediction, nor provide a reliable confidence measure associate to them. Furthermore, the architecture they propose has key differences with ours, while they propose a nested hierarchy in a end-to-end fashion (i.e., features associated to the coarse and fine levels are shared from the top to the bottom of the network), we enforce sequential information bottleneck. As we show in the following sections this sequence of coarse to fine low dimensional representations facilitate a robust calibration and combination of nested outputs. Yan et al. (2015) study the problem of nested learning for two nested levels of granularity, and optimize for a final fine prediction. Their design is specific for a two-level category hierarchy while our work generalize to any number of nested levels, and we simultaneously can train and test up to an arbitrary level of granularity. Another important difference is that the focus of their work is on the implementation details and performance of their two-hierarchy network versus traditional end-to-end learning. As they mention in the conclusion of their work, future work should aim to extend their ideas to more than two hierarchical levels and to contextualize their empirical results into a theoretical framework. Our work takes steps in these two specific directions. This is also a fundamental difference with the works developed by Triguero & Vens (2016) and Wehrmann et al. (2018). Moreover, all these approaches suppose that every sample is annotated for all the granularity levels. In contrast with them, we can efficiently train (and predict) our model on dataset that provide only coarse labels, intermediate levels, or fine labels. Our approach is therefore more general in that regard. Also, we show that if testing conditions shift from the ones on training, we can still provide relatively confident coarser labels while avoiding overconfident (erroneous) fine predictions (see, e.g., Figure 1, right; and Figure 5 in the supplementary material). Also, the questions of neural network overconfidence and output readability are never addressed in those works, which is crucial for both results interpretation, and output combination. Finally, because our solution can leverage information from datasets with different granularity, we are able to analyze how different proportions of training coarse and fine data affects models cost, performance and robustness.

## 3 NESTED LEARNING

Let us assume we want to classify the popular hand written digits of MNIST LeCun & Cortes (2010). An input image can be represented as a realization $x$ of the random variable $X$.[2] We denote as $\mathcal{X}$ the alphabet of $X$. Associated to $x$, there is a *label $y$* that corresponds to the actual number the person writing the character wanted to represent. The label $y$ is a realization of the random variable $Y$. In this illustrative case, $Y$ can take 10 different values: $\mathcal{Y} = \{0, 1, ..., 9\}$. Of course $Y$ and $X$ are not independent random variables. Generally, $Y$ precedes $X$, and the problem of classification can be stated as the problem of inferring $y$ from an observed sample $x$, i.e., $Y \to X \to \hat{Y}$. $\hat{Y}$ denotes a new random variable (estimated from $X$) which *approximate $Y$*. More precisely, a common practice is to find a mapping $X \to \hat{Y}$ such that the probability $P(\hat{Y} \neq Y | X)$ is minimized.

**Nested classification.** In this work we focus on the inherent hierarchical structure most classification problems have. For example, imagine now that we have hand written characters including numbers, lower case letters, and capital letters. It would be intuitive to first attempt to classify these characters into three categories: numbers, lower case letters, and capital letters. Then, depending on this coarse

---

[2]Capital letters will be used to denote random variables and lower case letters to denote the value of a particular realization.

classification we can perform a finer classification, i.e., classifying the numbers into $0 - 9$ classes, the letters into $a - z$, and so forth.

Of course, we could have an arbitrary number of nested random variables associated to different levels of labels granularity. Here, subscripts indicate the granularity of the label, for example, $Y_{i-1}$ is the closest coarse level of $Y_i$. $Y_i^k$ represents the random variable associated to each $k$ value in the closest coarse node, i.e., $Y_i^k$ represents $Y_i$ given that $y_{i-1} = k$, $k \in \mathcal{Y}_{i-1}$.

**Definition 3.1.** We define $Y_1, ..., Y_n$ as a discrete sequence of nested labels if $H(Y_i|Y_{i+1}) = 0$ $\forall i \in [1, n-1]$. $H$ denotes the standard definition of entropy for discrete random variables.

**Definition 3.2.** A discrete sequence of nested labels $Y_1, ..., Y_n$ is strictly nested if $H(Y_i|Y_{i-1}) < H(Y_i)$ $\forall i \in [2, n]$.

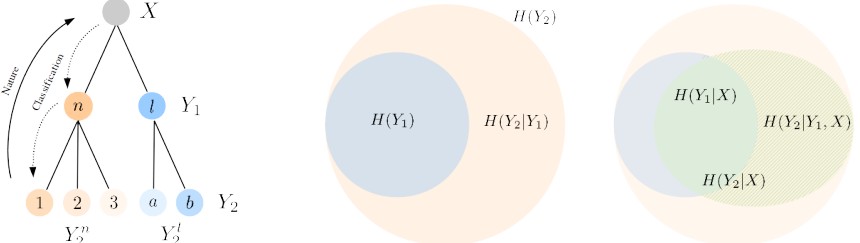

Figure 2: On the left we illustrate the taxonomy of an example of strictly nested labels. First handwritten characters are classified as "number" or "letter," and then these categories are refined into specific numbers and letters. $X$, $Y_1$ and $Y_2$ denote random variables representing the input, the coarse label, and the fine label respectively. $Y_i^k$ represents the random variable associated to each value $k$ in the coarser node, i.e., $Y_i$ given that $y_{i-1} = k$, $k \in \mathcal{Y}_{i-1}$. The diagram in the center, illustrate the entropy of a fine and coarse level, and how having information about a coarser level may reduce the entropy of the fine level. The right diagram illustrates the reduction of uncertainty on the labels given the input, and how the uncertainty on the fine labels can be reduced even further if input information and coarse information are combined.

The core of this work is to formulate classification problems such that the information of the input is extracted in a hierarchical way. To this end, intermediate (coarse) predictions $\hat{Y}_i$ are jointly learned. The two key components of the proposed approach are: (i) nested information bottlenecks and (ii) a combination of the predicted coarse and fine labels. The analysis provided in this section is agnostic to most of the implementation details (which are addressed in coming sections).

**Hierarchical information bottlenecks and the role of skipped connections.** We assume the random variable $X$ contains information about the sequence of strictly nested labels $Y_i$, i.e., $H(Y_i|X) < H(Y_i)$, as illustrated in Figure 2. More precisely, $H(Y_i|X) > H(Y_{i-1}|X)$ and $H(Y_i|X) < H(Y_i|Y_{i-1}, X)$. To exploit these properties, we use standard DNN layers (convolutional, pooling, normalization, and activation layers). As illustrated in Figure 3, we begin by guiding the network to find a low dimensional feature representation $f_1$ such that $H(f_1(X)) \ll H(X)$ while, $I(f_1(X), Y_1)$ is *close* to $I(X, Y_1)$ where $I(\cdot, \cdot)$ stands for the standard mutual information. (DNNs are remarkably efficient at compressing and extracting the mutual information between high dimensional inputs and target labels [Shwartz-Ziv & Tishby (2017), Moshkovitz & Tishby (2017)].)

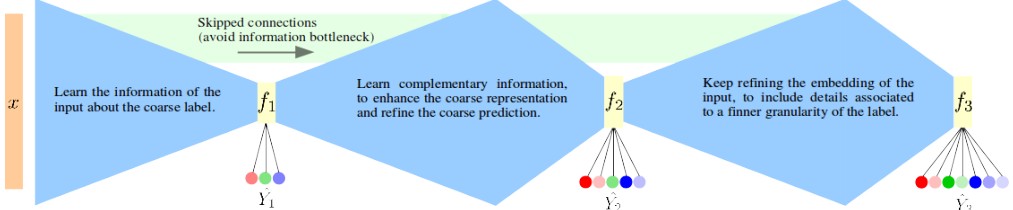

Figure 3: Illustrative scheme of the proposed framework. From left to right, the input data $x \sim X$, a first set of layers that extract from $X$ a feature representation $f_1$, which leads to $\hat{Y}_1$ (estimation of the coarse label $Y_1$). $f_1$ is then jointly exploited in addition with complementary information of the input. This leads to a second representation $f_2$ from which a finer classification is obtained. The same idea is repeated until the fine level of classification is achieved. It is important to highlight that this high level description of the proposed model can be implemented in multiple ways. In the following sections we present our own implementation and provide details of the specific architecture we chose for experimental validation.

The second step consists in learning the *complementary* information such that when combined with the representation $f_1$, allows us to achieve a second representation $f_2$ from which the second hierarchical label $Y_2$ can be inferred. To this end, skipped connections play a critical role as will be further detailed next. As we discussed before,

$$I(X, Y_i) = H(X) - \underbrace{H(X|Y_i)}_{>H(X|Y_{i+1})} < I(X, Y_{i+1}). \tag{1}$$

On the other hand, we want each feature embedding $f_i$ to compress the information of $X$ while $I(f_i(X), Y_i) \approx I(X, Y_i)$. Equation 1 means that the finer the classification the more information from $X$ we need. Notice that while in many DNNs, skipped connections have proved to help on the networks compactness and to mitigate vanishing gradients Szegedy et al. (2017), in the present work they are included for a more fundamental reason. (See the experiments presented in the supplementary material Section F.2 for complementary empirical evidence and validation.) If we do not consider skipped connections, $X \to f_i(X) \to f_{i+1}(X)$ forms a Markov chain where $I(X, f_{i+1}(X)) \leq I(X, f_i(X))$ (data-processing inequality) which contradicts Equation 1. Section 4 presents experiments illustrating the impact of skipped connections on nested learning, and complements the discussion started here.

**Combination of nested outputs.** We will present in the following sections experimental evidence showing that nested learning leads to an improvement in performance and robustness. In addition, the explicit combination of nested predictions, since the network simultaneously produces all the outputs $Y_i$ (with corresponding confidence), can improve the accuracy and robustness even further.

Our motivation is to explicitly refine the fine prediction leveraging the information of all the coarser outputs, i.e., $\{\hat{Y}_1, ..., \hat{Y}_i\} \to \tilde{Y}_i$. Let us define $s_i(q)$ the network output score associated to the event $Y_i = q$. In general, if $s_i(q) > s_i(w)$ most likely $P(Y_i = q) > P(Y_i = w)$, but $P(Y_i = q) \neq s_i(q)$. In other words, a score value of $0.3$ does not mean the sample belongs to this class with $30\%$ probability. This can be addressed by calibrating the outputs, which consists of mapping output scores into an estimation of the class probability $s_i(q) \to P_{\hat{Y}_i}(q)$, as defined and thoroughly explained in Zadrozny & Elkan (2002). $P_{\hat{Y}_i}(q)$ denotes the calibrated output of the network which approximates $P(Y_i = q)$. (We will precise how calibration is performed in the following section.) Then, we can use the estimated probability associated to a fine label $P_{\hat{Y}_i}$ to compute the conditional probability $P(Y_i = y_i | Y_{i-1} = k)$. This is achieved by re-normalizing the finer labels associated to the same coarse label, i.e.,

$$P_{\hat{Y}_i | \hat{Y}_{i-1}}(q) = \frac{P_{\hat{Y}_i}(q)}{\sum_{w \in \mathcal{Y}_i^{k_q}} P_{\hat{Y}_i}(w)}, \tag{2}$$

where $\mathcal{Y}_i^{k_q}$ denotes the set of labels at granularity level $i$ that share with $q$ the same coarser label $k_q$. Finally, the estimated conditional probability is combined with the prior of the coarser prediction to recompute the fine prediction $P'_{\hat{Y}_i}(q) = P_{\hat{Y}_i | \hat{Y}_{i-1}}(q) P_{\hat{Y}_{i-1}}(k_q)$.

### 3.1 IMPLEMENTATION CHALLENGES AND DETAILS

**Training.** Let $G_{\theta, \eta}(x) = (f_i(x, (\theta_j)_{j=1,...,i}), g_i(f_i, \eta_i))_{i=1,...,m}$ be the function coded by our network, where $m$ denotes the number of granularity levels and as before $x$ an input sample. Each sub-function $g_i$ corresponds to the output of granularity $i$ (computed from the feature bottleneck $f_i$). $G$ depends on parameters $(\theta_j)_{j=1,...,i}$ which are common to the sub-functions of coarser granularities, and some granularity-specific parameters $\eta_i$. The general framework of the architecture follows Figure 3, meaning a trunk of convolutionnal filters with parameters $\theta$ and fully connected layers for each intermediate outputs with parameters $\eta$.

Training this kind of model with a disparity of samples per granularity is hard, and naively sampling random batches of training data leads to a noisy gradient computation Kokkinos (2017). (See the supplementary material Section F.3 for complementary experimental validation and analysis.) In order to overcome this issue, we organize the training samples and train the network in a cascaded manner. The dataset $\mathcal{D}$ is organized in subsets of samples labeled up to granularity $i$ for $i = 1, .., m$. Formally we can write $\mathcal{D} = (\mathbf{x}, \mathbf{y})$ with $\mathbf{x}$ the set of inputs and $\mathbf{y}$ the set of labels. We consider that $\mathbf{x} = (\mathbf{x_i})_{\mathbf{i=1},..,\mathbf{m}}$ and $\mathbf{y} = (\mathbf{y_i})_{\mathbf{i=1},..,\mathbf{m}}$ where $\mathcal{D}_i = (\mathbf{x_i}, \mathbf{y_i})$ represents the subset of data for which the label is known up to the granularity level $i$.

Having the dataset partitioned in this fashion naturally leads to a cascaded training of the network. We train the model to solve a sequence of optimization problems using $(\mathbf{x_i}, \mathbf{y_i})$ as the training examples at each step. We can write this sequence $(P_i)$ as: $(P_i) : \min_{(\theta_j, \eta_j)_{j=1,...,i}} \sum_{j=1}^{i} \alpha_j \mathcal{L}_{n_j}(\hat{Y}_j, Y_j)$, where $\mathcal{L}_n$ is the $n$-categorical cross-entropy and $\alpha$ are the weights for each individual loss. We first start to train the network on coarse labels and we gradually add finer labels and start optimizing deeper parameters in an iterative way. We support our cascaded training protocol against a more standard one with experimental results presented in the supplementary (Section F.3). (Additional details are presented as supplementary material Section B.)

**Scores calibration.** As shown by Hein et al. (2018), deep neural networks with ReLU activations tend to produce overconfident results for samples which are out of the training distribution. In order to mitigate this and have meaningful outputs that can be combined, we consider a two step calibration method. The first step consists in adding a "rejection" class for each level of granularity. Synthetic samples associated to this class are generated from an uniform distribution. By training the network with this supplementary class we mitigate the problem of overconfidence for sample that are far from the training distribution. However, keeping a fixed coverage on the input space would require a number of samples that grows exponentially with the dimension of the input space.

To overcome this problem, we think of the network as a sequence of encoders in the latent spaces $g_i(f_i(x))$ (defined as the penultimate layer prior each prediction), and fine-tune the last layer adding synthetic *out-of-distribution* samples from an uniform distribution in the latent space. The latent space has a much smaller dimension than the original input space, and therefore is tractable to synthesise samples with a uniform coverage of the latent space. (Additional details are provided in the supplementary material Section B.)

The second calibration step is a classic temperature scaling introduced in Guo et al. (2017). This technique consists in scaling the output of the fully-connected layer before the softmax activation by an optimal temperature parameter. Given $x$ the input data and $g$ the function coded by a neural network before the softmax activation $\sigma$, the new calibrated output is given by $\bar{g} = \sigma(\frac{g}{T})$. The temperature parameter $T$ is tuned so that the mean confidence of the predictions matches the empirical accuracy, more precisely, we want to minimize, $\mathbb{E}\left[ \left| P(\hat{Y} = Y | \hat{p} = p) - p \right| \right]$, whereas before, $\hat{Y}$ denotes the network prediction of $Y$, and $\hat{p}$ is the empirical confidence associated to it. The previous expression can be approximated, using a validation partition of the data, by computing the Expected Calibration Error (ECE) $ECE = \sum_{j=1}^{N} \frac{|B_j|}{n} |acc(B_j) - conf(B_j)|$, Naeini et al. (2015). This measure takes the weighted average between the accuracy and confidence on $N$ bins $B_j$, $j = 1...N$. $n$ denotes the total number of samples, and $|B_j|$ the number of samples on the bin $B_j$.

## 4 EXPERIMENTS AND DISCUSSION

We consider three sets of publicly available datasets for experimental evaluation: the handwritten digits from MNIST (LeCun & Cortes (2010)), the small clothes images from Fashion-MNIST (Xiao et al. (2017)), and CIFAR10 (Krizhevsky et al.). First we study how the actual nested nature of the labels affects nested learning. Then we compare end-to-end training versus the proposed nested approach. In a third group of experiments we evaluate different combination methods to leverage all the predictions into a refined fine prediction. Then we evaluate the impact of skipped connections. Finally, we study how the ratio of coarse and fine labels affects the network's performance. Additional experiments are presented in the supplementary material, section A, F

**Hierarchical structure versus random grouping.** We group the MNIST samples into two groups of nested labels: "VG" and "RG," both with three labels of granularity. VG corresponds to a hierarchy that groups visual similarities; the coarse class groups digits into $\{\{3, 8, 5, 0, 6\}, \{9, 4, 7, 1, 2\}\}$, the intermediate class into $\{\{3, 8, 5\}, \{0, 6\}, \{9, 4, 7\}, \{1, 2\}\}$, and of course the fine class into $\{0\} - \{9\}$. Additional information on how we grouped the labels for the three datasets is presented in Appendix C. RG consists of a 3 level (coarse/middle/fine) grouping based on the order of $\mathbb{N}$. Empirical results shows that grouping the labels based on visual similarities leads to better results in terms of accuracy, intuitively supporting the idea of nested learning. (See the results presented in Table 4 provided in the supplementary materials.)

**Standard end-to-end training versus nested learning.** Let us define $|\mathcal{D}_i|$ as the number of training samples that are annotated up to the level of granularity $i$. To understand if adding more samples with

coarse annotation helps improving the performance on the fine task, we compared models trained exclusively with fine data $\mathcal{D}_A = \mathcal{D}_3$ and models trained fine data plus coarsely annotated data $\mathcal{D}_B = \mathcal{D}_3 + \mathcal{D}_2 + \mathcal{D}_1$. Table 1 shows the accuracy for the fine, middle, and coarse outputs for the standard end-to-end network versus the same architecture trained using nested learning and additional coarse and middle data. If we compare the lines for which $\mathcal{D}_3 = 20\%$, we observe that adding coarse and middle granularity samples improves the accuracy even for the fine task. Moreover, it also improves the robustness of the model. We tested this models when test data is distorted and shifted from the conditions at training. Distortion 1 to 4 correspond to four levels (increasing the severity) of "turbulence-like" image distortion, the implementation of this distortion is inspired on the work of Meinhardt-Llopis & Micheli (2014) (details are provided in the supplementary material, Section D). Again looking at Table 1 we see that the model trained with additional coarse and middle samples is more robust and less overconfident. A similar pattern is observed for MNIST and fashion-MNIST datasets (see for example, tables 5, and 6 in the supplementary material). Complementing these results, Section F.1 in the supplementary material compared the proposed nested architecture with a Multi-Task learning approach.

| method | Original | Distortion 1 | Distortion 2 | Distortion 3 | Distortion 4 | Bound on Std |
|---|---|---|---|---|---|---|
| Coarse (end-to-end), $\mathcal{D}_3 = 20\%$ | 96.0 / 97.6 | 87.0 / 94.0 | 82.5 / 93.4 | 80.2 / 92.9 | 77.0 / 93.0 | |
| Coarse (end-to-end), $\mathcal{D}_3 = 32\%$ | **96.8** / 98.2 | 86.2 / 93.9 | 82.1 / 93.1 | 78.3 / 92.7 | 75.1 / 92.9 | ±0.3 |
| Coarse (nested) Ours, $\mathcal{D}_{1,2,3} = 20\%$ | 96.5 / 96.7 | **87.8** / 92.5 | **84.9** / 91.5 | **81.4** / 90.9 | **78.2** / 90.6 | |
| Middle (end-to-end), $\mathcal{D}_3 = 20\%$ | 84.1 / 89.8 | 65.2 / 79.6 | 56.7 / 76.0 | 48.9 / 74.9 | 42.6 / 75.9 | |
| Middle (end-to-end), $\mathcal{D}_3 = 32\%$ | **87.5** / 92.8 | **65.5** / 81.5 | 56.3 / 79.0 | 47.8 / 78.7 | 41.3 / 79.4 | ±0.4 |
| Middle (nested) Ours, $\mathcal{D}_{1,2,3} = 20\%$ | 85.2 / 85.0 | 65.4 / 73.5 | **58.1** / 70.4 | **50.3** / 69.5 | **43.9** / 69.7 | |
| Fine (end-to-end), $\mathcal{D}_3 = 20\%$ | 75.9 / 88.4 | 50.3 / 67.9 | 42.8 / 64.9 | 34.2 / 65.5 | 28.4 / 73.4 | |
| Fine (end-to-end), $\mathcal{D}_3 = 32\%$ | **81.0** / 88.4 | **52.0** / 73.3 | 41.8 / 71.4 | 32.8 / 71.9 | 26.8 / 73.4 | ±0.5 |
| Fine (nested) Ours, $\mathcal{D}_{1,2,3} = 20\%$ | 77.4 / 77.7 | 51.6 / 62.6 | **43.2** / 59.8 | **34.7** / 57.7 | **29.1** / 57.7 | |

Table 1: Accuracy and mean confidence ($Acc\%/Conf\%$) for Cifar10 dataset. Coarse, fine, and middle indicate the accuracy at each level of the label. End-to-end, denotes the model trained with exclusively data annotated for the fine label. We compare two end-to-end models trained with different amounts of data with fine labels. We first set $\mathcal{D}_3 = 20\%$, which we increase afterwards to $\mathcal{D}_3 = 32\%$. On the other hand, "nested" denotes the same architecture, trained with coarse middle and fine labeled data. In this experiment we set $\mathcal{D}_1 = \mathcal{D}_2 = \mathcal{D}_3 = 20\%$. We repeated the distortion generation 10 times (for all the levels of distortion), the last column "Bound on Std" shows the maximum standard deviation obtained across "Distortion 1" to "Distortion 4," therefore the value on the last column can be interpreted as a bound on the variability of the reported results. (Similar results are reported for MNIST and fashion-MNIST datasets, see tables 5, and 6 in the supplementary material.)

The previous discussion is interesting as it shows that including additional coarse data tends to help also the discrimination of the fine task. However one may argue that the comparison is unfair, as one model sees more data than the other. That is a very interesting point and we address it in following experiments where we study how the proportion of fine and coarse granularity data affect performance (for a fixed budget and different cost models). For now, let us observe what happens if we increase the amount of fine data from $\mathcal{D}_3 = 20\%$ to $32\%$ (which assuming a linear cost model equals the budget of training with $\mathcal{D}_1 = \mathcal{D}_2 = \mathcal{D}_3 = 20\%$). As expected, (see Table 1) the performance on clean test data improves for the end-to-end model. However, (see columns Distortion 1-4) it generalizes less to unseen (distorted) test data and also becomes significantly more overconfident.

**Output combination.** Kuncheva (2004) presented many useful combination methods for both one-hot encoding classifiers and continuous scores. We compare some of those combination methods (e.g., Mean, Product, and Majority Vote) to the one that we designed specifically for multiple nested outputs. (Details and numerical results are provided in the supplementary material, Section E and Table 3.) We observed that different combination methods perform similar on test data that matches the train data, while the proposed method outperforms the others when test samples are distorted.

**The role of skipped connections.** Skipped connections are included in order to allow information flow from the input to the finer feature bottlenecks. Table 2 shows the accuracy for two networks with the same structure, trained on the same data, one including skipped connections (SC) and the other-one not. Section F.2 in the supplementary material complements these results, and provides empirical measurements of the mutual information between different components of the model with and without skipped connections.

| Method | Original | Distortion 1 | Distortion 2 | Distortion 3 | Distortion 4 |
|---|---|---|---|---|---|
| Coarse (without SC) | 99.6 | 96.3 | 91.3 | 85.3 | 79.2 |
| Coarse (with SC) Ours | **99.7** | **97.2** | **93.2** | **87.7** | **81.4** |
| Middle (without SC) | 99.3 | 92.1 | 79.4 | 66.8 | 58.4 |
| Middle (with SC) Ours | **99.5** | **95.1** | **87.3** | **79.2** | **65.5** |
| Fine (without SC) | 98.9 | 88.2 | 72.4 | 56.4 | 44.0 |
| Fine (with SC) Ours | **99.2** | **94.2** | **84.9** | **69.5** | **53.2** |

Table 2: Comparison of the same network structure, trained on the same coarse, middle, and fine data, with and without skipped connections. 20% of fine middle and coarse samples of MNIST dataset where selected for training. As in the previous experiments, Distortion 1-4 correspond to test distorted samples with turbulence-like distortion (described in the supplementary material).

**Working on a budget.** As we discussed before, establishing a fair comparison between models trained using more fine or coarse data is not trivial. To address this problem, we investigated three cost models: linear, convex, and concave. For the linear model, we assume the cost associated to annotate one sample $(x, y_i)$ is proportional to $|\mathcal{Y}_i|$. Analogously, for the convex/concave cost model, we assume the cost of annotating a sample $(x, y_i)$ is proportional to $g(|\mathcal{Y}_i|)$ with $g(\cdot)$ being strictly convex/concave. For each budget and cost model, we created hundreds of train sets with different amounts of coarse, middle, and fine samples. Figure 4 shows the accuracy on the MNIST test set for different budgets and cost models. It is interesting to observe that for a convex cost model, increasing the number of coarse annotations produces better results for the same budget; while as expected, the opposite is observed for a concave cost model. We present additional experiments in the Figure 10 in the supplementary material.

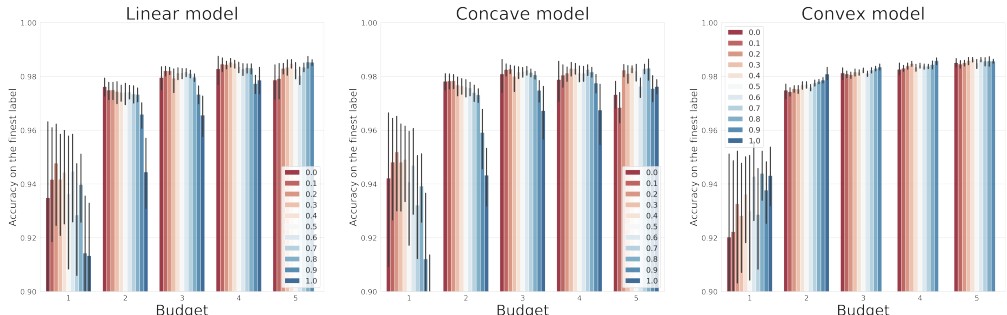

Figure 4: Accuracy on the MNIST data when different ratios of coarse, middle, and fine samples are selected for training. Plots report the accuracy on the prediction of the finest category (ten classes: $0 - 9$). If we define $n_1$, $n_2$ and $n_3$ the number of samples for which we know only the coarse, middle, and fine label respectively, the budget associated to a training set is $B_{n_1, n_2, n_3} = n_1 g(|\mathcal{Y}_1|) + n_2 g(|\mathcal{Y}_2|) + n_3 g(|\mathcal{Y}_3|)$. $g$ represents a cost function associated to labeling a coarse, middle, and fine sample. In this experiment, we tested three models for $g$, a linear model where the cost is linear, a concave model (we chose $g(u) \propto log(u)$), and a convex model (we chose $g(u) \propto e^u$). The colors represent the proportion of coarse samples in the training set. Blue represents more proportion of coarse samples, and red a larger proportion of fine labels. (Additional results are presented in Figure 10 in the supplementary material.)

## 5 CONCLUSION

In this work we introduced the concept of *nested learning*, which improves classification accuracy and robustness. Moreover, it allows to leverage information from datasets annotated with different levels of granularity. Additionally, experiments suggest that nested models have a very desired behaviour, e.g., they gradually break as the quality of the test data deteriorates. We showed that implementing nested learning using a hierarchy of information bottlenecks provides a natural framework to also enforce calibrated outputs, where each level comes with its confidence value.

Given a fixed budget and model, we studied what is the economically most efficient strategy for labeling data. Furthermore, experimental results show that if the amount of fine training samples is constant, then adding samples with only a coarse annotation increases the performance and robustness for the fine task. To recap, the introduced nested learning framework performs as expected from our own human experience, where for good data we can provide high level inference with high confidence; and when the data is not so good, we can still provide with high confidence some level of inference on it.

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

# Supplementary Material

## A    ADDITIONAL OUTPUT EXAMPLES

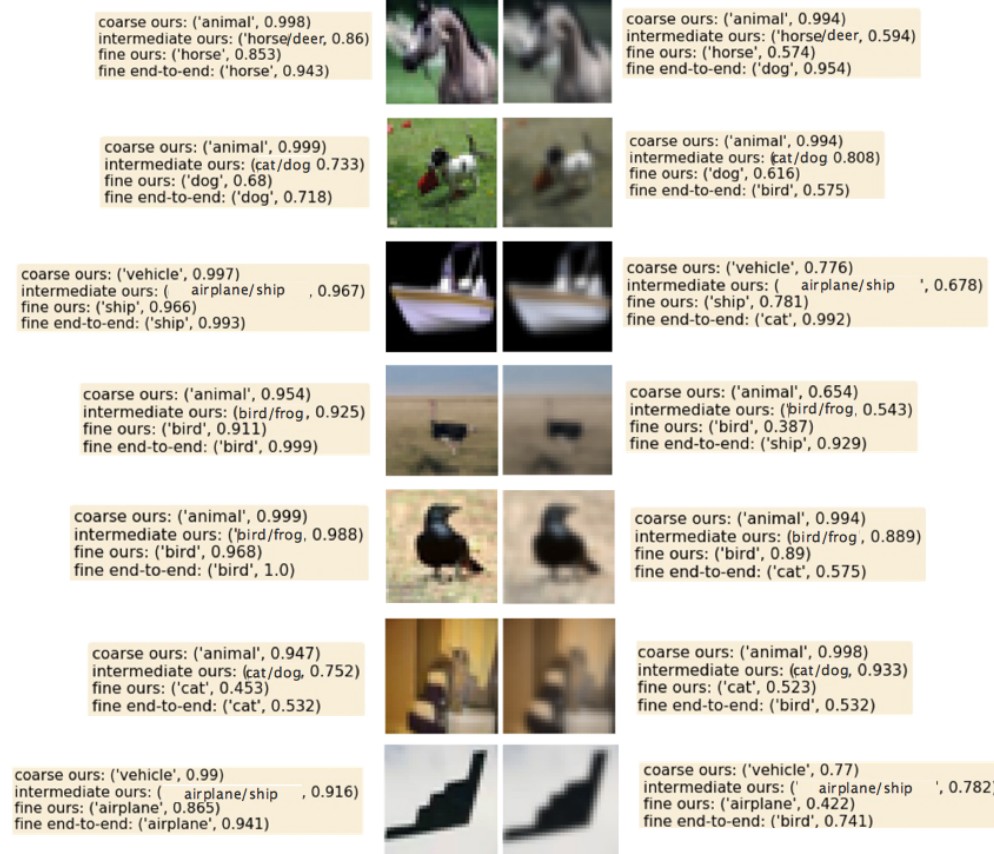

Figure 5: These results complement the example illustrated in Figure 1. The output of a standard (end-to-end) DNN and our proposed nested learning version are compared. On the left we show clean images from the test set of CIFAR-10 dataset, on the right, the same examples but blured. Next to each image our prediction (for the fine, middle, and coarse level) and the prediction of a standard (end-to-end) DNN are displayed. Both DNN share the same architecture and their performance is compared on Table 1 (rows corresponding to "(end-to-end) $\mathcal{D}_3 = 32\%$" and "(nested) our $\mathcal{D}_{1,2,3} = 20\%$"). As shown in Table 1 the performance of both networks is similar on clean data (i.e., data that match the train distribution), but our approach can provide more accurate middle and coarse predictions when the input data is corrupted, moreover, we are significantly less overconfident on the prediction of out-of-distribution test samples.

## B    IMPLEMENTATION DETAILS

### B.1    ARCHITECTURE

The architecture of our model is presented in Figure 6. We evaluate the same architecture for MNIST and Fashion-MNIST as the images have the same size and number of channels. For CIFAR10, the architecture of our model is very similar but with an increased depth of the convolutional filters. Classifying images from CIFAR10 is indeed a harder problem than classifying images from MNIST or Fashion-MNIST, and therefore, it requires a model with more parameters. The architecture we use is an adaptation of the U-Net, designed to fit our proposed framework of nested information

bottlenecks. The U-Net is indeed a good starting point, as it meets most of the criteria that we presented in Section 3. First, it consists of a convolutional network that enforces a bottleneck representation. Second, it presents skip connections that allow information of the input to flow to deeper components of the network.

We also added Batch-Normalization (BN) layers after every convolutional filters in order to achieve a faster convergence. BN is very helpful to mitigate the vanishing gradient phenomenon. Also, since it introduces randomness during training, BN acts as a regularization. Classic L1 regularization was shown to be similar to BN in Luo et al. (2018), while including BN layers also improves training speed and stability as explained by Santurkar et al. (2018).

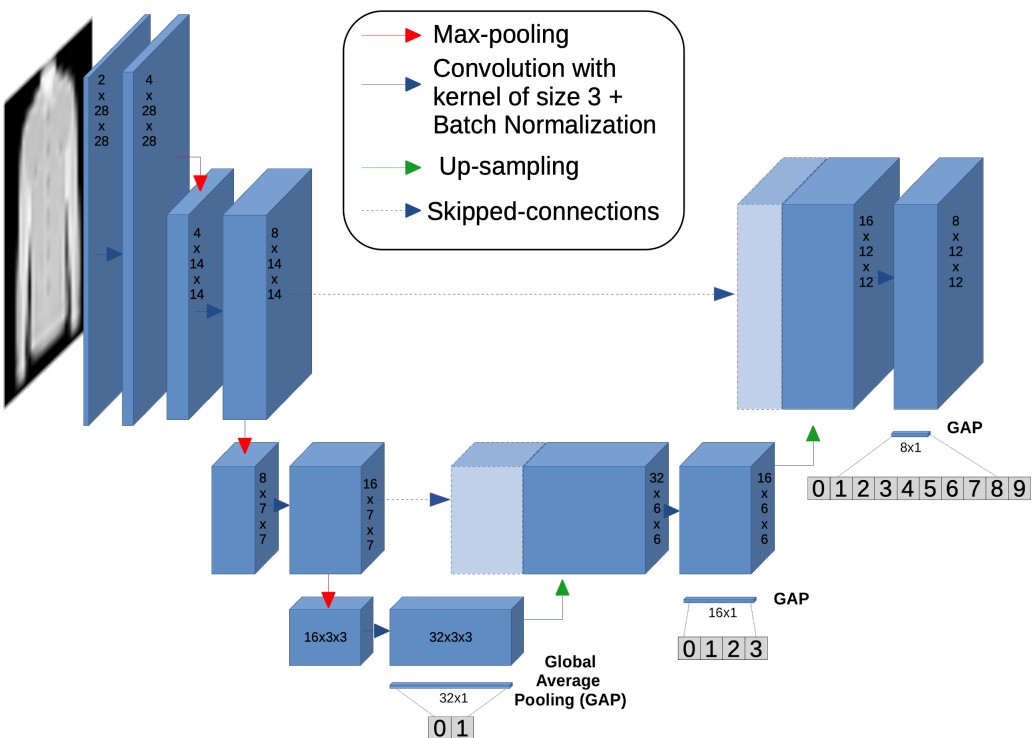

Figure 6: Architecture of our CNN for both Fashion-MNIST and MNIST. This model is an adaptation of the U-Net network (Ronneberger et al. (2015)) designed to fit our nested learning framework. The blue boxes represent the feature extracted by convolutional layers. We perform a global average pooling rather than a flattening to decrease the number of parameters. After the global average pooling, the feature vector is normalized with respect to the L2 norm (instance normalization layer). The normalized features are followed by fully connected layers to compute the final output. The model used to test CIFAR10 set is very similar but has convolutional layers with more kernels handle this (sightly more complex) task.

## B.2 Training

We train the models in an iterative way. First we optimize the weights for the coarse prediction with the samples that are only coarsely annotated and freeze the remaining weights. Then, we optimize the weights up to the intermediate output with the samples that are coarsely and intermediately annotated. Finally we train the whole network, with the samples that are coarsely, intermediately, and finely annotated. (In general, this process has as many steps as levels of granularity.) Each training step is performed using ADAM optimizer with different learning rates: $2 \times 10^{-3}$ for the first step, $1 \times 10^{-3}$ for the second, and $5 \times 10^{-4}$ for the last. We stop the training of each step when stagnation of the validation loss is observed.

### B.3 CALIBRATION

The calibration consists of two main steps. First a "rejection" class is modeled using an uniform distribution on the latent space. This models out-of-distribution samples and mitigates overconfident predictions on portions of the feature space where no training data is observed. The second step consists of temperature scaling to convert output scores into approximations of class probabilities.

**The "rejection" class.** For every level of granularity $i$ and for every sample of the training dataset, we store the normalized outputs of the global averaging layer (GAP) in a dataset $D_i$. The samples have size $s_i$ and are normalized with respect to the L2 norm, therefore, they live in $\mathcal{B}^{s_i}(0,1)$, the unitary sphere in $\mathbb{R}^{s_i}$ centered in zero. We randomly sample $n_i$ new instances from an uniform distribution in $\mathcal{B}^{s_i}(0,1)$. These samples (associated to a new "rejection" class) are aggregated to $D_i$ and the fully connected layers fine tuned.[3] Naturally, the larger $|D_i|$ and $s_i$, the larger $n_i$ should be. We set $n_i \propto |D_i| \times \mathcal{S}(s_i)$ where $\mathcal{S}(s_i)$ is the area of the hypersphere of unitary radius in the $s_i$-dimensional space. Figure 7 illustrates for a one dimensional toy example, how the proposed ideas provide and efficient solution to reduce outputs overconfidence on out-of-distribution input samples.

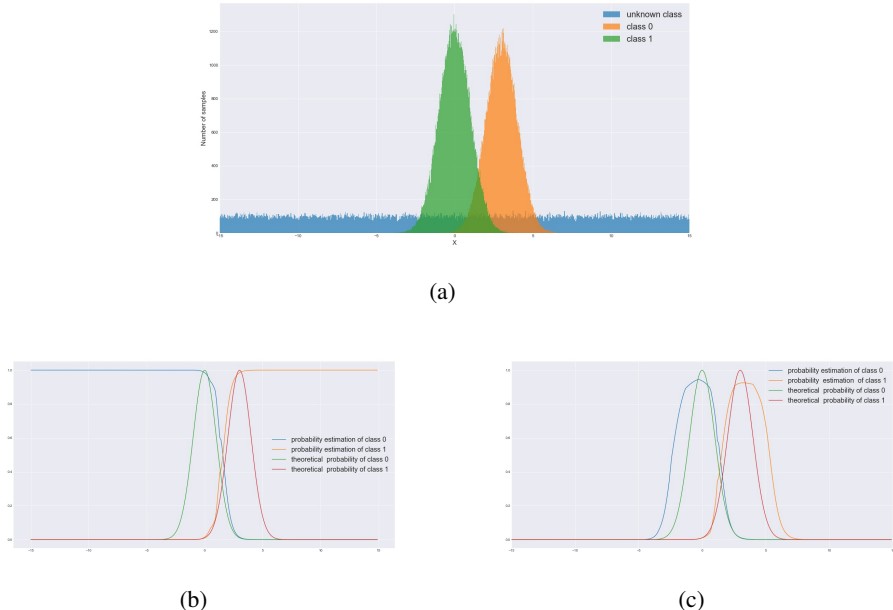

(a)

(b)                                                    (c)

Figure 7: One dimensional toy example that illustrates the problem of prediction overconfidence for input samples that are far from the training distribution. In this example, two classes ("1" and "0") are considered and we assume that the input $x \sim X$ is one-dimensional. (a) illustrates the empirical distribution of each class (on green $P(X|Y = 1)$ and yellow $P(X|Y = 0)$). In addition, we illustrate (blue distribution) the uniform distribution from which we sample synthetic training instances associates the "rejection" class. Figure (b) shows the confidence output associated to the class "1" and "0" for different values of $x$, for a model trained only on the original data (standard approach). Figure (c) illustrates the output of the same DNN trained with the samples associated to the classes "1" and "0", plus the synthetic samples from the "rejection" class.

**Temperature scaling.** Temperature scaling improves calibration by converting arbitrary score values into an approximation of class probabilities. As explained in Section 3.1, we minimize the empirical ECE metric over the validation set, composed of 5000 unseen samples for all the dataset we tested our method on. To find the optimal temperature $T$, we compute the ECE over 50 values between 1 and 3 and select the one for which the ECE is minimized. If $T = 1$ the model is already well calibrated. On all our experiments, the minimal ECE value was always reached for $T$ values strictly lower than 3.

---

[3]To this end, we used ADAM optimizer with a learning rate of $10^{-3}$.

## C  LABEL GROUPING

In Section 4, we showed that nested learning performs better if the taxonomy of the labels has a real nested structure (e.g., one based on visual similarity). To group fine labels into a meaningful nested structure, we first train a shallow neural network and classify images into the fine classes. Then we used the confusion matrix $M$ associated to this auxiliary classifier to establish which classes are closer to each other.

For MNIST and Fashion-MNIST for example, we wanted to group the labels in 2 coarse categories which also contained 2 intermediate categories. To this end, we find the 10-permutation $l$ applied to both the rows and columns of $M$, such that the non-diagonal $5 \times 5$ matrices of $M$ had the lowest possible L1-norm. We iterate this process to find the intermediate categories. It is computationally hard to go through all the permutations, therefore, we follow the ideas proposed by Behrisch et al. (2016) to perform matrix reordering with a reduced complexity.

Figure 8 presents the groups of labels we obtained for MNIST, Fashion-MNIST, and CIFAR10. Our results show natural and intuitive taxonomies, see, e.g., how MNIST digits are grouped according to a natural shape-oriented similarity, with 3 and 8 in the same intermediate class for example.

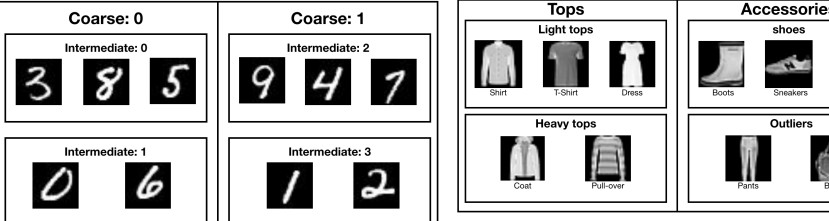

(a) Taxonomy obtained for MNIST          (b) Taxonomy obtained for Fashion-MNIST

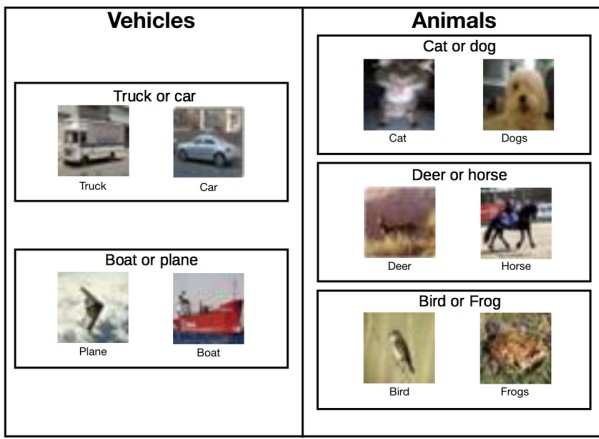

(c) Taxonomy obtained for CIFAR10

Figure 8: Nested groups of labels obtained by minimizing the non-diagonal components on the confusion matrix of an auxiliary simple classifier.

## D  PERTURBATIONS

Selecting realistic and meaningful perturbations to test DNN models is a non trivial problem. For example, adding Gaussian noise mainly affects the high frequency components of the input images and we observed that both standard (end-to-end) and nested networks were not severely affected by this type of perturbation. In this work we focus on structural deformations inspired by a model of turbulence. The pseudo-code of this perturbation is presented in Algorithm 1 and was inspired in

the work of Meinhardt-Llopis & Micheli (2014). Figure 9 illustrates the distortion of an example image from MNIST dataset for different levels of *turbulence intensity*.

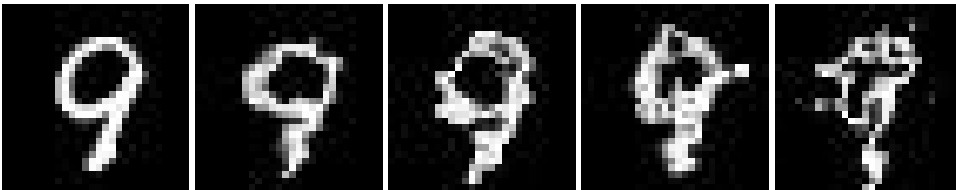

Figure 9: Example of different perturbations applied during testing time. From left to right: the original sample, and the distorted version with parameters $(S, T) = (1, 0.8)$, $(S, T) = (1, 1.0)$, $(S, T) = (1, 1.3)$, and $(S, T) = (1, 1.5)$ respectively.

---

**Algorithm 1:** Turbulence distortion

---

**Data:** the input: $I \in \mathbb{R}^{w,h}$, the parameters $(S, T) \in \mathbb{R}^2$
**Result:** the distorted image $I_{dist}^{(S,T)}$
Creating a vector field (u,v) for the distortion:
$u, v = normal\_noise((w, h)), normal\_noise((w, h))$
$u, v = gaussian\_filter(u, S), gaussian\_filter(v, S)$
$u, v = u \times \frac{T}{std(u)}, v \times \frac{T}{std(v)}$
Interpolate the image with the obtained vector field:
$I_{dist}^{(S,T)} = bilinear\_interpolate(I, u, v)$

---

## E    COMBINATION METHODS

Combining multiple classifiers is a standard approach in machine learning. However, most approaches combine similar outputs and are not designed for the specific problem of nested learning. We compared some standard combination methods and our calibration-based combination strategy.

Combination methods can be classified into two categories. First those that combine or vote discrete classification results, where classifier outputs are considered as one-hot encoding vectors. For example, the *Majority Vote (MA)* which consists in aggregating the decision of multiple classifiers and selecting the candidate that receives more votes. The second category of methods combine classifiers continuous outputs. As we show in the following, the second class of methods are more suitable for the combination of nested outputs.

For example, suppose that given an input sample $x$, the classifier outputs an estimations of the probability associated to each coarse, middle, and fine label: $P_{\hat{Y}_1}$, $P_{\hat{Y}_2}$, and $P_{\hat{Y}_3}$. In order to apply standard combination methods, we need to transform the coarse and intermediate predictions into a fine prediction and vice-versa. To transform the probability associated to a coarse label into a finer granularity, we can assume the finer classes associated to each coarse class are equally probable, i.e., the probability associated to a coarse node, is divided equally into the finer nodes associated to this label. On the other hand, to aggregate probabilities associates to a finer level into a coarser level, we can simply add those probabilities associated to the same coarse node.

Once probabilities associated to coarse levels are propagated to the fine levels and vice-versa, we can combine them using the *mean* or the *product* rule as described in Kuncheva (2004). Table 3 shows the result of combining nested outputs with standard combination methods and our strategy described in Section 3.1.

## F    ADDITIONAL EXPERIMENTS

| Distortion | Without comb. | Ours | Coarse & Fine | Mean | Product | MV |
|---|---|---|---|---|---|---|
| Original | **98.8** | 98.6 | 98.7 | 98.7 | 98.7 | 98.5 |
| Distortion 2 | 80.8 | **82.8** | 82.1 | 82.2 | 82.4 | 80.1 |
| Distortion 4 | 45.8 | **50.6** | 49.6 | 48.6 | 49.6 | 47.5 |

Table 3: Comparison of the fine accuracy for different combination techniques. The model is trained on MNIST dataset with $\mathcal{D}_1 = \mathcal{D}_2 = \mathcal{D}_3 = 10\%$

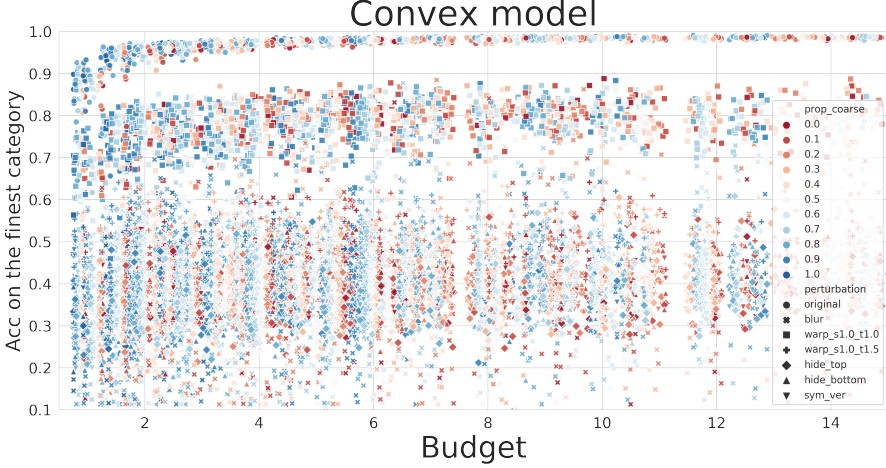

Figure 10: Accuracy on the classification of the fine label for MNIST data. These results complement the results presented in Figure 4. Figure 4 illustrates the results on clean test data grouping five levels of budgets and for three cost models (linear, concave, and convex). In this figure, we show the performance of each individual model. For each model we display the accuracy on the clean data (dots) as well as the accuracy on different types of distortions. As before, the proportion of coarse and fine labels during training is illustrated coloring each data point, blue indicates a higher proportion of coarse samples while red a higher proportion of fine samples.

| Perturbation | Original |
|---|---|
| VG coarse | **99.7** |
| RG coarse | 99.3 |
| VG middle | **99.5** |
| RG middle | 98.9 |
| VG fine | **98.6** |
| RG fine | 98.5 |

Table 4: Comparison of the visual label grouping (VG) and random label grouping (RG) in terms of accuracy for our model trained on MNIST.

| method | Original | Distortion 1 | Distortion 2 | Distortion 3 | Distortion 4 |
|---|---|---|---|---|---|
| Coarse (end-to-end), $\mathcal{D}_3 = 32\%$ | 98.9 / 99.3 | 90.7 / 96.4 | 87.5 / 95.9 | 85.1 / 95.5 | 84.0 / 94.7 |
| Coarse (end-to-end), $\mathcal{D}_3 = 20\%$ | 99.0 / 99.3 | 94.5 / 96.9 | 90.1 / 96.1 | 87.3 / 94.9 | **85.9** / 94.5 |
| Coarse (nested) Ours, $\mathcal{D}_{1,2,3} = 20\%$ | **99.2** / 99.4 | **96.2** / 97.6 | **93.1** / 96.6 | **89.3** / 96.0 | 84.7 / 95.5 |
| Intermediate (end-to-end), $\mathcal{D}_3 = 32\%$ | **94.3** / 95.8 | 82.1 / 90.9 | 77.2 / 89.5 | 73.0 / 88.2 | 69.6 / 87.1 |
| Intermediate (end-to-end), $\mathcal{D}_3 = 20\%$ | 93.7 / 95.4 | 84.3 / 91.7 | 79.5 / 90.2 | 74.0 / 89.0 | 71.3 / 88.3 |
| Intermediate (nested) Ours, $\mathcal{D}_{1,2,3} = 20\%$ | 94.2 / 94.7 | **87.3** / 91.2 | **82.6** / 89.3 | **77.7** / 88.5 | **72.8** / 87.3 |
| Fine (end-to-end), $\mathcal{D}_3 = 32\%$ | **88.3** / 91.5 | 70.8 / 83.5 | 62.6 / 81.3 | 55.0 / 80.5 | 48.3 / 79.6 |
| Fine (end-to-end), $\mathcal{D}_3 = 20\%$ | 87.1 / 91.1 | 70.1 / 84.7 | 62.3 / 83.0 | 54.7 / 81.1 | 50.3 / 80.2 |
| Fine (nested) Ours, $\mathcal{D}_{1,2,3} = 20\%$ | 86.7 / 87.9 | **73.7** / 81.3 | **67.6** / 77.6 | **57.6** / 76.4 | **51.2** / 74.4 |

Table 5: Accuracy and mean confidence ($Acc\%/Conf\%$) for Fashion-MNIST dataset. Coarse, fine, and middle indicate the accuracy at each level of the label. End-to-end, denotes the model trained with exclusively data annotated for the fine label. We compare two end-to-end models trained with different amounts of data with fine labels. We first set $\mathcal{D}_3 = 20\%$, which we increase afterwards to $\mathcal{D}_3 = 32\%$. On the other hand, "nested" denotes the same architecture, trained with coarse middle and fine labeled data. In this experiment we set $\mathcal{D}_1 = \mathcal{D}_2 = \mathcal{D}_3 = 20\%$.

| Method | Original | Distortion 1 | Distortion 2 | Distortion 3 | Distortion 4 |
|---|---|---|---|---|---|
| Coarse (end-to-end), $\mathcal{D}_3 = 16\%$ | 99.4 / 99.5 | 96.1 / 97.8 | 90.9 / 95.3 | 82.3 /92.6 | 72.0 / 90.9 |
| Coarse (nested) Ours, $\mathcal{D}_{1,2,3} = 10\%$ | **99.5** / 99.5 | **96.9** / 97.5 | **92.3** / 95.4 | **85.9** / 93.8 | **79.6** / 92.1 |
| Intermediate (end-to-end), $\mathcal{D}_3 = 16\%$ | 99.1 / 99.2 | 94.8 / 96.9 | 87.5 / 93.3 | 74.6 / 89.8 | 60.7 / 88.0 |
| Intermediate (nested) Ours, $\mathcal{D}_{1,2,3} = 10\%$ | **99.2** / 99.2 | **95.5** / 96.0 | **87.9** / 92.6 | **77.0** / 89.4 | **66.5** / 87.9 |
| Fine(end-to-end), $\mathcal{D}_3 = 16\%$ | 98.5 / 98.7 | 91.9 / 95.4 | 80.4 / 90.9 | 67.2 / 85.7 | 49.7 / 83.1 |
| Fine (nested) Ours, $\mathcal{D}_{1,2,3} = 10\%$ | **98.6** / 98.4 | **92.8** / 93.5 | **82.8** / 87.8 | **67.5** / 83.0 | **50.6** / 80.4 |

Table 6: Accuracy and mean confidence ($Acc\%/Conf\%$) for the MNIST dataset. Coarse, fine, and middle indicate the accuracy at each level of the label. End-to-end, denotes the model trained with exclusively data annotated for the fine label. On the other hand, "nested" denotes the same architecture, trained with coarse middle and fine labeled data. In this experiment we set $\mathcal{D}_1 = \mathcal{D}_2 = \mathcal{D}_3 = 20\%$.

## F.1 COMPARING WITH A COMMON MTL ARCHITECTURE

Classic Multi-Task Learning (MTL) DNN approaches extract common features that are on the top the architecture branched for the prediction of multiple tasks. To this end, a common MTL architecture consists of shared convolutionnal blocks followed by task-specific classification (fully connected) layers. Figure 11 illustrates an MTL architecture designed to obtain a nested classification of MNIST characters. The network enforces an information bottleneck that encodes the input in a (64,1) feature vector. Then, it is connected to the three classification branches implemented by a sequence of fully-connected layers. This MTL model and our nested model have approximately the same number of parameters and are trained in an identical way on MNIST dataset.

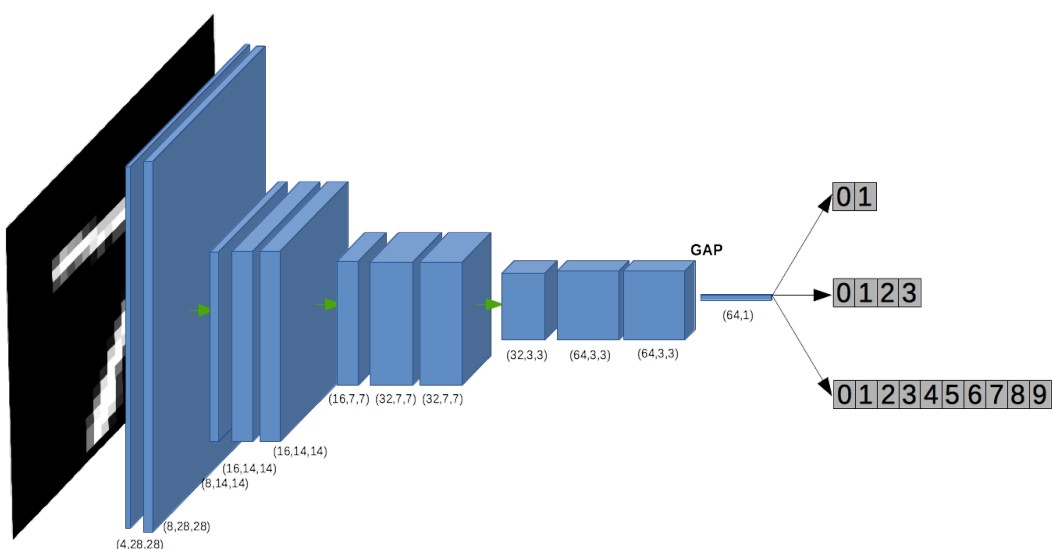

Figure 11: Architecture of a test Multi-Task-Learning CNN. This architecture corresponds to a sequence of three blocks of two (convolutional + batch normalization) layers followed by a Maxpooling Layer. In order to reduce the dimension of the penultimate feature vector, we perform a global average pooling. This network and our Nested CNN have approximately the same number of parameters and are trained using the same training protocol and data.

Table 7 shows the classification accuracy for our nested architecture presented in Section B and the multi-task learning architecture presented above. On the original distribution both architectures perform roughly equivalently, but on perturbed test data the proposed nested model is considerably superior. Nested learning outperforms its MTL counterpart with a gap of 18% for the fine task and a gap of more than 10% for the coarse task for distortion levels 3 and 4. These results provide additional evidence that having a hierarchical nested architecture promotes robustness and leads to more consistent models.

| Method | Original | Distortion 1 | Distortion 2 | Distortion 3 | Distortion 4 |
|---|---|---|---|---|---|
| Coarse (MTL),$\mathcal{D}_{1,2,3} = 10\%$ | 99.5 / 99.7 | 92.4 / 96.5 | 82.6 / 94.3 | 73.8 /92.9 | 66.1 / 93.1 |
| Coarse (nested) Ours, $\mathcal{D}_{1,2,3} = 10\%$ | **99.5** / 99.5 | **96.9** / 97.5 | **92.3** / 95.4 | **85.9** / 93.8 | **79.6** / 92.1 |
| Intermediate (MTL), $\mathcal{D}_{1,2,3} = 10\%$ | 99.1 / 99.4 | 91.1 / 94.9 | 75.5 / 90.8 | 60.2 / 88.8 | 49.9 / 88.2 |
| Intermediate (nested) Ours, $\mathcal{D}_{1,2,3} = 10\%$ | **99.2** / 99.2 | **95.5** / 96.0 | **87.9** / 92.6 | **77.0** / 89.4 | **66.5** / 87.9 |
| Fine(MTL), $\mathcal{D}_{1,2,3} = 10\%$ | 98.9 / 99.0 | 91.0 / 89.9 | 70.9 / 81.3 | 48.6 / 77.9 | 32.4/ 80.1 |
| Fine (nested) Ours, $\mathcal{D}_{1,2,3} = 10\%$ | 98.6 / 98.4 | **92.8** / 93.5 | **82.8** / 87.8 | **67.5** / 83.0 | **50.6** / 80.4 |

Table 7: Classification accuracy and mean confidence ($Acc\%/Conf\%$) for an example of MTL and nested learning on the MNIST dataset. Coarse, fine, and middle indicate the accuracy at each level of the label. MTL denotes the standard Multi-task Learning architecture described in Section F.1. On the other hand, "nested" denotes our nested model which enforces hierarchical feature embeddings. In this experiment we set $\mathcal{D}_1 = \mathcal{D}_2 = \mathcal{D}_3 = 20\%$.

### F.2 SKIPPED CONNECTIONS AND THEIR IMPACT ON THE FLOW OF INFORMATION

Obtaining an empirical measure of the mutual information (MI) between two high dimensional random variables is a very hard numerical problem (Paninski (2003)). However, recent progresses in deep learning made a numerical approximation (MINE) tractable by exploiting the flexibility of neural networks and properties of the mutual information. Belghazi et al. (2018a) proved that a MI estimation can be seen as an optimization problem. They rely on the following characterization of the mutual information as the Kullback-Leibler (KL-) divergence,

$$I(X, Z) = D_{KL}(\mathbb{P}_{XZ} || \mathbb{P}_X \otimes \mathbb{P}_Z). \tag{3}$$

Based on this, the authors use the Donsker-Varadhan representation of the KL divergence, which introduces a dual optimization problem,

$$D_{KL}(\mathbb{P}||\mathbb{Q}) = \sup_{T:\Omega \to \mathbb{R}} \mathbb{E}_\mathbb{P}[T] - \log(\mathbb{E}_\mathbb{Q}[e^T]), \tag{4}$$

where the supremum is taken over all the functions $T$ such that the two expectations are finite.

Since the parameters of a neural network can be used to encode a large space of functions, the authors propose to solve Equation equation 4 for $T_\theta \in \mathcal{F}$, where $\mathcal{F}$ denotes the space of functions encoded by a pre-defined network architecture with parameters $\theta \in \Theta$. An approximation of the MI can be obtained solving the problem

$$I_\Theta(X, Z) = \sup_{\theta \in \Theta} \mathbb{E}_{\mathbb{P}_{XZ}}[T_\theta] - \log(\mathbb{E}_\mathbb{Q}[e^{T_\theta}]) \tag{5}$$

Equation 5 can be numerically solved using standard optimization tools of deep learning (see Belghazi et al. (2018b) for details).

In order to measure the impact of skipped-connections from an MI perspective, we measure the empirical approximation of the mutual information between different sections of the proposed nested architecture. We compared these results for the same network with and without skipped connections. Let us refer to the network with skipped connections with the subscript 1 and the network without skipped connections with the subscript 2. We define different variables of interest at particular stages of the network and estimate the mutual information between them. We will call $F_1(X)$ and $F_2(X)$ the variable corresponding to the feature map created after the 2nd Maxpooling layer (see Figure 11) with or without skipped connections. Similarly, let $G_1(X)$ and $G_2(X)$ be the feature maps obtained before the second GAP layer. We will also consider $H_1(X)$ and $H_2(X)$ the coarsest feature maps obtained before the first GAP. (Before feeding these features to MINE algorithm we performed average pooling to reduce the dimension of the input variables.)

We estimated the MI between $I(F_i(X), G_i(X))$ and $I(F_i(X), H_i(X))$, $i = 1$ meaning "with skipped connections" and $i = 2$ "without skipped connections" (the network without skipped connections is re-trained to allow the model to adapt to this new configuration). Figure 12 (left side) sketches in which sections of the model we are measuring the mutual information, and (right side) the evolution of the MI estimation (MINE algorithm) for $3k$ steps. After convergence of MINE algorithm we compared the mutual information between the random variables $F_i$, $H_i$ and $G_i$ with and without skipped connections (illustrated in green in the sketch of the left side of figure 12),

$$\Delta_1 = I(F_1(X), G_1(X)) - I(F_1(X), H_1(X)) = 0.37, \tag{6}$$
$$\Delta_2 = I(F_2(X), G_2(X)) - I(F_2(X), H_2(X)) = 0.21 \approx \Delta_1/2. \tag{7}$$

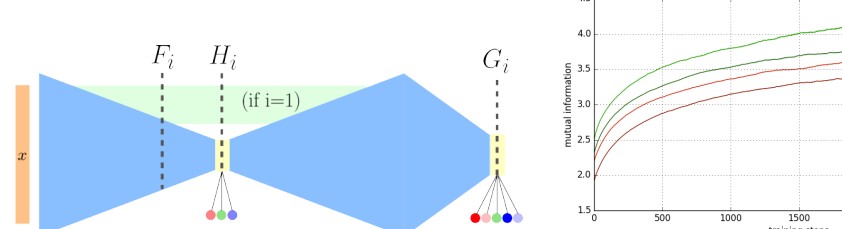

Figure 12: Left: we sketch the sections of the model where the mutual information is compared. Right: evolution of the estimated MI for $I(F_1(X), G_1(X))$, $I(F_2(X), G_2(X))$, $I(F_1(X), H_1(X))$, and $I(F_2(X), H_2(X))$ during the optimization of the MINE estimator. The MINE network is composed of three fully connected hidden layers of 100 neurons each and elu activations.

As shown in Equation 6 skipped connections play an important role to allow information to flow to the feature representation of the finer classes. This provides additional numerical evidence to the discussion presented in Section 3 supporting the importance of including skipped connections and avoiding information bottleneck at the finer classification.

### F.3 COMPARISON OF TRAINING PROTOCOLS

Our training protocol is organized in a cascaded way. As explained in Section 3.1, the reason for this is two-fold. First, this would eliminate the problem of the noisy stochastic gradient computations which happens when we there is not enough samples in the mini-batch for a specific task/granularity. Second, it provides a good initialisation of the weights associated to the coarser levels, and we observed empirically that this leads to a faster convergence and better accuracy. In order to illustrate the impact of the training protocol, we compare the results obtained by training the exact same model, with the same data, but using two different training strategies. We compared the described cascade methodology with the standard methodology (i.e., selecting batches of random samples out of the training set). As samples are annotated heterogeneously (with all or some fine to coarse annotations) we defined the unified loss in each mini-batch as

$$\mathcal{L}_{batch} = \sum_{j=1}^{M} \sum_{i=1}^{N} \alpha_i \times \omega_{ij} \mathcal{L}_i(y_{ij}, \hat{y_{ij}}),$$

$M$ denotes the number of samples in the batch, $N$ the number of nested level, where $\omega_{ij}$ is a masking vector which indicates if the sample $j$ is annotated or not for the level of granularity $i$, $\alpha_i$ are global weight that help to balance the loss associated to each level $\mathcal{L}_i$.

We observed that cascaded training achieves substantially better performances than the traditional training as we can see in Table 8. The gaps are very significant also on the training distribution as we can observe in the first column. Additionally, if we studied the behaviour of the network while, e.g., see the results presented in Figure 13. As we can see, the proposed protocol is more suitable for the proposed multi-level problem leading to faster convergence and better models. After 26 epochs of coarse learning, we add the intermediate data for 30 epochs before adding the fine data for 44 more epochs. We can observe that very few epochs after the introduction of the intermediate data, the accuracy of the intermediate classifier is already superior to a standard training procedure. We observe this phenomenon again for the fine classification, see for example, the training evolution over the green area illustrated in Figure 13. In addition we can observe that the accuracy of the cascaded training classifiers oscillates far less than the accuracy of the traditionally trained classifiers. This provides additional evidence that our method helps to deal with the noisiness of the stochastic gradient estimation.

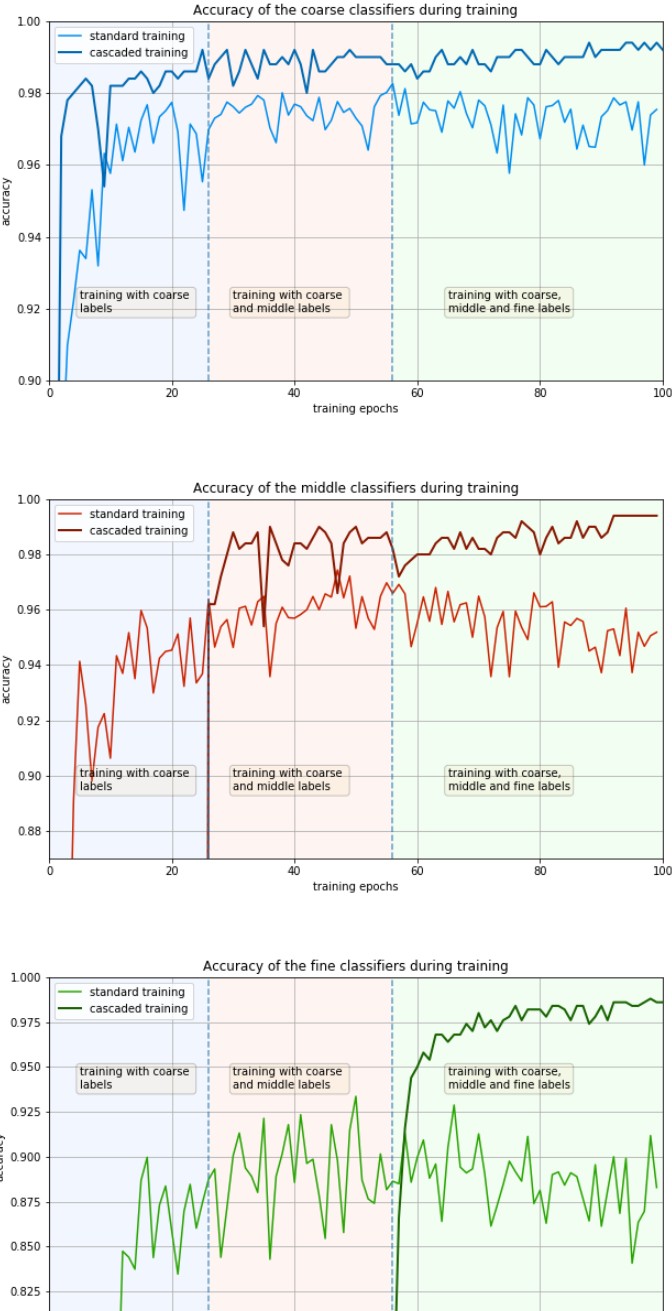

Figure 13: These plots compare the accuracy of the classifiers for the cascaded training and the traditional training. On the top, we represent the evolution of the coarse classifiers during training. In the middle, we represent the evolition of the accuracy of the intermediate classifiers. At the bottom, we represent the evolution of the accuracy of the fine classifiers.

| Method | Original | Distortion 1 | Distortion 2 | Distortion 3 | Distortion 4 |
|---|---|---|---|---|---|
| Coarse (traditional training),$\mathcal{D}_{1,2,3}=10\%$ | 98.8 / 82.0 | 94.9 / 75.4 | 88.4 / 72.5 | 80.2 /71.0 | 73.4 / 70.8 |
| Coarse (cascaded training), $\mathcal{D}_{1,2,3}=10\%$ | **99.5** / 99.5 | **96.9** / 97.5 | **92.3** / 95.4 | **85.9** / 93.8 | **79.6** / 92.1 |
| Intermediate (traditional training), $\mathcal{D}_{1,2,3}=10\%$ | 96.9 / 81.2 | 91.1 / 74.2 | 81.1 / 69.5 | 66.8 / 67.6 | 53.5 / 69.3 |
| Intermediate (cascaded training), $\mathcal{D}_{1,2,3}=10\%$ | **99.2** / 99.2 | **95.5** / 96.0 | **87.9** / 92.6 | **77.0** / 89.4 | **66.5** / 87.9 |
| Fine (traditional training), $\mathcal{D}_{1,2,3}=10\%$ | 94.2 / 84.1 | 86.7 / 79. | 73.4 / 71.9 | 53.2 / 68.8 | 37.6/ 70.5 |
| Fine (cascaded training), $\mathcal{D}_{1,2,3}=10\%$ | **98.6** / 98.4 | **92.8** / 93.5 | **82.8** / 87.8 | **67.5** / 83.0 | **50.6** / 80.4 |

Table 8: Accuracy and mean confidence ($Acc\%/Conf\%$) for the MNIST dataset. Coarse, fine, and middle indicate the accuracy at each level of the label. In this table we compare the results of a traditional training to a cascaded training on the same architecture. In this experiment we set $\mathcal{D}_1=\mathcal{D}_2=\mathcal{D}_3=20\%$. We notice significant improvement with our method.

