# OpenReview forum: "NESTED LEARNING FOR MULTI-GRANULAR TASKS"
_ICLR.cc/2020/Conference — Reject_

### Official Review · AnonReviewer2 · 2019-10-23
**Official Blind Review #2**

**Rating:** 3

**Review:**

Nested learning for multi-granular tasks

1. Summary
The paper considers a framework for classification with a hierarchy of labels [from coarse to fine]. The paper proposes a network architecture with multiple bottleneck layers, one for each label level, and skip connections. The objective function is the standard classification loss. The experiments show that coarse labels help learning and can improve label efficiency, i.e. don’t need all fine labels to get good classification performance.

2. Opinion and rationales

Whilst I think the execution of ideas is good and the motivation is very practical, I’m leaning towards “reject” for this paper due to the reasons below. I welcome the authors’ clarification and am willing to reconsider my view.

i. The paper justifies the proposed architecture as successive information compression into label embeddings using some entropy based criteria (as in information bottleneck literature). This ensures the relationship of the entropies between the label layers. However, I’m not sure this justification is necessary (albeit being a valid one) given that the training/later sections do not come back to this justification.

ii. The novelty of the proposed architecture and training approach is low. The network is a nested structure of successive classifiers. The proposed calibration using rejection class and temperature scaling is not new.

iii. it would be better if the proposed architecture + method are validated on more real-world datasets with a more natural label grouping scheme, and compared to alternative architectures, e.g. multi-task learning with a shared network except the output layer.

3. Minor details

Some citations should be enclosed in brackets

**Experience Assessment:**

I have read many papers in this area.

**Review Assessment: Checking Correctness Of Derivations And Theory:**

I assessed the sensibility of the derivations and theory.

**Review Assessment: Checking Correctness Of Experiments:**

I assessed the sensibility of the experiments.

**Review Assessment: Thoroughness In Paper Reading:**

I read the paper at least twice and used my best judgement in assessing the paper.

---

> ### Author Response · Authors · 2019-11-15
> **Answers to Review #2**
>
> We would like to start by sincerely thanking the reviewer for his/her time and dedication reviewing this paper. He/she presents interesting comments and questions that we address in the following.
>
> Q: The paper justifies the ...
> A: Following reviewers suggestion, we added three experiment to provide additional empirical evidence to the theoretical ideas presented in Section 3. (i) We compare the proposed nested architecture with a multi task learning (MTL) approach (see Section F.1 in the updated version of the supplementary material, and in particular the results shown in Table 7); (ii) We added experiments showing the impact of skipped connections in the amount of mutual information between features above and below an information bottlenecks (see supplementary material section F.2., Figure 12 and equations (6)-(7)); and (iii) We tested alternative training protocols as suggested by reviewer nr. 4 (Section F.3).
>
> The experiments show how the proposed nested learning provides more robust feature extraction for nested problems compared with its MTL counterpart. In addition we show that when skipped connections are included, more information flows to from the first layers to the features associated to finner classification. To this end, we used MINE algorithm (Belghazi et al. 2018) to obtain an empirical estimation of the mutual information of different features extracted in the network. The experiments provide additional empirical evidence that when skipped connections are included, more information flows from the first layers to the features associated to finer classification (approximately by a factor of two). This provides empirical support to the ideas discussed from a theoretical perspective in the Section 3 pag. 4.
>
> Q: The novelty of the proposed ...
> A: To make more clear the novelty of this work, we substantially modified Section 2 to make more explicit the connection and differences between our work previous literature. In particular we highlight the main differences between our work and the work that is closer to ours: Kim et al.(2018), Yan et al. (2015), Triguero & Vens (2016), Wehrmann et al. (2018). There are key differences between our work previous approaches (please see the updated version of the manuscript for a detailed discussion, in particular, the last paragraph of Section 2). To the best of our knowledge the important framework of nested training and testing has not been explicitly proposed before.
>
> Q: it would be better if the proposed ...
> A: Following the reviewer’s suggestion we compared the proposed architecture with a standard multi-task learning network. The results are presented in the updated version of the manuscript (Section F.1, Figure 11 and Table 7). The architecture of the MTL network is described in Figure 11. Both architectures (design to have roughly the same number of parameters) can be trained to achieve the simultaneous classification of nested labels, however as illustrates Table 7 (page 19) enforcing a nested feature representation significantly improves the robustness of the model.
>
> Q: Minor details Some citations should be enclosed in brackets
> A: We fixed this issue.

---

### Official Review · AnonReviewer1 · 2019-10-23
**Official Blind Review #1**

**Rating:** 3

**Review:**

I read the author response, thank you for responding to my questions.

Original review:

This paper presents a hierarchical learning approach that trains neural network classifiers on a known hierarchy of labels.  The experiments show that the approach makes a network more robust to distortion compared to standard end-to-end learning.  The approach in this paper and its formalization of the task are interesting, but the significance of the techniques is somewhat unclear and the experiments could be more thorough in terms of the baselines and data sets considered.

From the related work section, the relationship between this paper and the previous work is somewhat complicated -- it is hard for a reader not deeply familiar with these previous works to understand the unique contribution made in the submission, and assess why it is significant.  For example, in the last paragraph of that section, many of the distinctions drawn between the submitted work and previous methods seem minor (including a confidence measure for a certain prediction, for example) or seemingly subjective (about whether an operation with a previous method was “natural” or could be done “transparently”).  Making crisper, less ambiguous distinctions between this work and previous work would help.

Likewise, the experimental results here do not compare against any of the hierarchical learning approaches discussed in the related work section.  The results show that the paper’s approach is more robust to distortion compared to standard end-to-end learning.  However, I was unclear on why it was not appropriate to compare against the other hierarchical learning methods from previous work.  Also, if the claim is improved robustness, I feel that evaluating against adversarial training (Madry et al., ICLR 2017) or similar approaches is necessary to understand the practical relevance of these improvements.

Finally, experiments that consider larger hierarchies (here, the number of target classes tends to be small, CIFAR-10 and MNIST each have ten classes, meaning the hierarchies are not very rich) would help illustrate the potential power of the techniques.

Minor
I didn’t understand the following statement, and given that it’s a fairly bold claim I would rephrase it or explain it better in the paper body rather than referring the reader to the appendix:
“A standard DNN unknowingly uses low quality data also to train higher layers, even if there is no high level information in the data.”

I don’t understand what the right arrow operator on the top of page 5 means.  From earlier statements it seems that I(f_i(X), Y_i) -> I(X, Y_i) means that the left quantity is approximately equal to the right.  But I’m not sure why to use an arrow for that rather than an \approx symbol.  The arrow would seem to imply that the left approaches the right in the limit, but if that is what you mean you should tell us in the limit of what.  Later the arrow is used to represent links in a Markov chain, adding further confusion for me.

I think it would be helpful if before Equation 1, you mentioned this holds for strictly nested Y_i’s (since earlier in the paper, Y_i referred to more general things).

I assume the ECE is computed over held-out validation data (i.e., not training data)?  The paper should say this.

Page 7: lineal -> linear

**Experience Assessment:**

I have published one or two papers in this area.

**Review Assessment: Checking Correctness Of Derivations And Theory:**

I assessed the sensibility of the derivations and theory.

**Review Assessment: Checking Correctness Of Experiments:**

I assessed the sensibility of the experiments.

**Review Assessment: Thoroughness In Paper Reading:**

I read the paper thoroughly.

---

> ### Author Response · Authors · 2019-11-15
> **Answers to Review #1**
>
> We would like to start by sincerely thanking the reviewer for his/her time and dedication reviewing this paper. He/she presents interesting comments and questions that we address in the following.
>
> Q: From the related ....
> A: We reformulated the discussion of the related work and provided more crisper and precise details and comparisons between our formulation and previous work. This updated discussion is provided in the new version of the manuscript, last paragraph of Section 2, pag. 3.
>
> Q: Likewise, the experimental results ....
> A: The proposed work focus on achieving a nested representation capable of leveraging data labeled at different levels of granularity and at the same time, capable of producing multiple nested outputs. As we show, as a consequence we obtain more robust and versatile solutions, though we do not explicitly aim at improving robustness as adversarial training does. In that sense, we think the ideas here proposed are actually complementary to those in adversarial learning. Moreover, it would be a very interesting future work to see how the ideas proposed here and standard techniques of adversarial training can be combined to improve even further the robustness of nested learning. In addition, we added an experiment in the updated version of the manuscript comparing the proposed approach with multi-task learning (see Section F.1. in the updated version of the supplementary material). For example, as shown in the new set of experiments presented in Table 7, the proposed architecture leads to a more robust representation compared to MTL (for a network trained on the same data and with approximately the same number of parameters). Again, this suggests that learning feature representations in a nested fashion inherently encourages models robustness (even though we do not explicitly enforce it as in adversarial learning).
>
> Q: I didn’t understand ...
> A: We rephrase this claim and now it reads: " ... when heterogeneous data with different quality and granularity of annotations (as in the example illustrated in Figure 1  (left)) is provided for training, low quality samples with coarse labels can help us to understand the structure of the coarser distributions (person, under 50) while simultaneously data with finer labels can contribute to the coarse and fine tasks. " .
>
> Q: I don’t understand what the right arrow ...
> A: We fixed this issue and make the notation more clear.
>
> Q: I think it would be helpful ....
> A: The reviewer is correct, we added this information in the updated version of the manuscript (last paragraph of Section 3, pag. 5).
>
> Q: Page 7: lineal -> linear
> We fixed this.

---

### Official Review · AnonReviewer4 · 2019-11-01
**Official Blind Review #4**

**Rating:** 3

**Review:**

Summary: The problem addressed is how to train a DNN to learn a hierarchical representation of the input that has heterogeneously annotated labels at various levels of granularity. After training the network’s goal is to emit sequentially finer grained labels with corresponding confidence. The authors use information theory to propose a network typology of information bottlenecks with skip connections to achieve this nested learning problem.

Decision: weak reject.

Reason: The proposal of learning a hierarchical representation is not new. Nor is the architecture. I think the interesting points (that are not really flesh out as much, but appear to be in the auxiliary material - section B.2) is the training regime. I would’ve liked to have seen more of what the role of the training regime is on the outcomes and how the network’s gradient’s behave in different regimes. But in general, the paper is well organized and argued, although there is a little belaboring of ideas of entropy and mutual information only to use it to buttress a point that is made and left hanging. The paper also defines too many concepts for every proposition it wants to support, making it cognitively costly to follow.

Feedback:
Pleasure reading the paper. Few points of feedback:

- Perhaps show the network gradients to help understand the dynamics of the learning
- What do you think is the role of the training regime (section B.2) on the outcomes you are observing? Do you think it would be worth observing the effect of changing the learning regime (and the accompanying gradient) on the outcomes?
- Generalization to any number of nested labels is not demonstrated
- The empirical demonstration of contribution of skip connections is not too
- Corollary to above, what do you think would be the role of attention in the hierarchical representation learning?

Questions:
1- Not clear why complementarity is a necessary condition of learning (section 3, before definition 1). Take for example the vehicle, wheels, truck example in figure 1. Learning F2 (wheels) features isn’t conditioned on correctly learning F1 (vehicle). In fact, could it be satisfactory, to first order approximation, to assume that learning finer-grained features first (roundness of wheels) and then combining lower level features (in what you refer to as Markov chain) may result in equal if not better accuracies? Is it possible to test this?

2- Not clear why calibration of outputs (section “combination of nested outputs” in p 5) is an approximation of P(Y_i=q).

3- Its not clear to me why the proposed rejection calibration method as a way of handling overconfidence is the right approach. Why this solution? Why not use, for example, regularization instead?

4- Table 1 of results. As the distortions increases the relative improvements of the nested learner appears to increase in CIFAR-10 results. Can you demonstrate this is not an artifact of the distortion generation strategy and is indeed a stable observation?

5- why is the marginal accuracy improvements so much larger for going from a budget of 1 to 2 than 2 to 5 in all cost functions? Does this not refute the claim that the nested model “gradually breaks”?

**Experience Assessment:**

I do not know much about this area.

**Review Assessment: Checking Correctness Of Derivations And Theory:**

I carefully checked the derivations and theory.

**Review Assessment: Checking Correctness Of Experiments:**

I carefully checked the experiments.

**Review Assessment: Thoroughness In Paper Reading:**

I read the paper at least twice and used my best judgement in assessing the paper.

---

> ### Author Response · Authors · 2019-11-15
> **Answers to Review #4**
>
> We would like to start by sincerely thanking the reviewer for his/her time and dedication reviewing this paper. He/she presents interesting comments and questions that we address in the following.
>
> Q: Perhaps show ....
> A: We performed additional experiments following this suggestions (see Section F.3, Figures 13 and Table 8).
>
> Q: What do you think is the role ....
> A: Yes, it is definitely worth exploring different training regimes as they play an important role in the effectiveness and efficiency of the training of nested solutions. We included additional experiments in the updated version of the paper showing the impact on training of different training protocols in (see new Section F.3, and in particular, the experiments presented in Figure 13 and Table 8).
>
> Q: Generalization to any number of nested ....
> A: We do demonstrate that our ideas can be applied in practice with two and three nested levels. This is substantially more general than previous work that is developed exclusively for two nested levels, see e.g. (Yan et al. 2015). In addition as the proposed method is general we think testing it in problems with more hierarchies -when data becomes available- is going to be straightforward (our code is also already available online). From a theoretical perspective, we do develop all our ideas with a general number of nested labels in mind (in contrast with previous work). The proposed guidelines for the network architecture, the combination of the outputs, the training protocol and so forth are generalize to an arbitrary number of nested levels.
>
> Q: The empirical demonstration of skip connections ....
> A: The reviewer is absolutely right. We showed in the original submission the impact of  skipped connections in the overall performance of the network (page 8 Table 2), but we did not explicitly validate empirically the impact of skipped connections in the flow of information (as we argue in page 4 Section 3). In the updated version of the manuscript we added experiments showing the impact of skipped connections in the amount of mutual information between features above and below an information bottlenecks (see the updated appendix, Section F.2, and Figure 12).
>
> Q: Role of attention ...
> A: We think is an interesting idea we did not explore it in this work, we think it could be discussed in future work and merged with the ideas we present  as they are complementary.
>
> Q: Not clear why ....
> A: We believe the review is confused here due to a poor choice we made at defining the name of some classes (we corrected this issue in the updated version, and explain the confusion in the following).
> In the experiment shown in Figure 1, the nested classes F1:Vehicle, F2: Wheels, and F3: Truck mean:
> - F1: The object is a vehicle (we name this class "vehicle")
> - F2: The object is a vehicle with wheels (in opposition, for example, to a boat or a plane). We named this class "wheels" to group ground vehicles, but not because we are actually detect or classify wheels specifically.
> We hope this explanation makes the Figure 1 more clear, and evacuate the reviewers confusion.  We also changed the categories name in the paper to avoid this confusion in the future.
>
> Q: Not clear why .....
> A: The term "output calibration" is not introduced or defined by us, we followed the most common nomenclature convention (see, e.g., Zadronzny & Elkan (2002)), we clarified this in the updated version of the manuscript (pag 5.).
>
> Q: Why not use, for example, regularization ....
> The proposed method is certainly an effective way of mitigating the problem of prediction overconfidence and is complementary to regularization methods. Indeed, the models we implemented include layers of batch normalization with has been proven to act as a weight regularization (see Luo et al. (2018)). We discussed this in more detail in the updated version of the manuscript (section B.1).
>
> Q: Table 1 ....
> A: This is a very stable observation. Indeed, we obtained consistent results on several runs and under different conditions. To provide a quantitatively measure of the stability of the results reported, we repeated 10 times the experiments presented in Table 1 and included in the updated version a bound on the standard deviation of the results.
>
> Q:Why is the marginal accuracy ....
> A: We believe that has to do with the amount of data, budget 2 is exactly two times budget 1 and that explains the large improvement between this two levels. On the other hand, after a certain level of accuracy is achieved the effect of adding data start to saturate, and that is why between budget 2 and 4 the relative improvement is more modest.  In regards to the claim that the nested model "gradually breaks" an important distinction should be made. More precisely, by "gradually breaks" we mean that as the quality of the input data deteriorates, the prediction accuracy starts to gradually decrease from the finner to the coarser outputs (as illustrated in the examples in Figure 5)

---

### Decision · Program_Chairs · 2019-12-19

**Decision:**

Reject

**Comment:**

This paper proposes a model architecture and training procedure for multiple nested label sets of varying granularities and shows improvements in efficiency over simple baselines in the number of fine-grained training labels needed to reach a given level of performance.

Reviewers did not raise any serious concerns about the method that was presented, but they were also not convinced that it represented a sufficiently novel or impactful contribution to an open problem. Without any reviewer advocating for the paper, even after discussion, I have no choice but to recommend rejection.

I'm open to the possibility that there is substantial technical value here, but I think this work would be well served by more extensive comparisons and a potentially revamped motivation to try to make the case for it that value more directly.